# New Data on the Rhamnose-Binding Lectin from the Colonial Ascidian *Botryllus schlosseri*: Subcellular Distribution, Secretion Mode and Effects on the Cyclical Generation Change

**DOI:** 10.3390/md21030171

**Published:** 2023-03-08

**Authors:** Giacomo Bovo, Loriano Ballarin

**Affiliations:** Department of Biology, University of Padova, 35121 Padova, Italy

**Keywords:** colonial ascidians, *Botryllus schlosseri*, rhamnose-binding lectin, phagocytes, secretion, blastogenic cycle

## Abstract

*Botryllus schlosseri* in a cosmopolitan ascidian, considered a reliable model organism for studies on the evolution of the immune system. *B. schlosseri* rhamnose-binding lectin (BsRBL) is synthesised by circulating phagocytes and behaves as an opsonin by interacting with foreign cells or particles and acting as a molecular bridge between them and the phagocyte surface. Although described in previous works, many aspects and roles of this lectin in *Botryllus* biology remain unknown. Here, we studied the subcellular distribution of BsRBL during immune responses using light and electron microscopy. In addition, following the hints from extant data, suggesting a possible role of BsRBL in the process of cyclical generation change or takeover, we investigated the effects of interfering with this protein, by injecting a specific antibody in the colonial circulation, starting one day before the generation change. Results confirm the requirement of the lectin for a correct generation change and open new queries on the roles of this lectin in *Botryllus* biology.

## 1. Introduction

Lectins are non-enzymatic and non-immunoglobulin carbohydrate-binding proteins that are widely diffuse in living organisms. They have been grouped into many families based on their carbohydrate recognition domains (CRDs). Animal lectins are characterised by different structures, carbohydrate specificities and physiological roles. Galectins, C-type lectins, P-type lectins, I-type lectins, fucolectins and pentraxins are some of the main families of animal lectins [1,2,3,4,5]. They modulate various physiological processes, such as cell–cell interactions during development and differentiation [1,6] or the recognition of nonself molecules by the immune system [5,7]. Various lectins have been described in tunicates, particularly in ascidians, mostly involved in immune recognition [8,9].

The family of rhamnose-binding lectins (RBLs) includes proteins that have multiple biological functions, being involved in fertilisation, cell proliferation, cytotoxicity and innate immunity [10,11,12,13]. Most of the RBLs have been described in fish where they are present in serum, embryos and eggs [11,12,13,14,15,16,17]. However, an increasing number of RBLs have been identified and characterised in invertebrates where they play important role in immunomodulation [18,19,20,21,22].

As far as ascidians are concerned, we identified a putative RBL from the solitary species *Ciona robusta* [3,23]. In addition, we isolated and studied an additional RBL, named *Botryllus schlosseri* rhamnose-binding lectin (BsRBL), in the colonial species *Botryllus schlosseri* [24,25].

In *B. schlosseri* colonies, zooids are grouped in star-shaped systems kept at the same developmental stage by the common colonial circulation that connects all the individuals (zooids and buds). Colonies grow through asexual reproduction by palleal budding. A cyclical generation change occurs, lasting 24–36 h, during which zooids are gradually resorbed in a process called takeover (TO) and replaced by their buds. A blastogenic cycle can be defined as the period between two successive TOs: it lasts a week at 20 °C [26]. In a previous work, we demonstrated that BsRBL is secreted by circulating phagocytes and the expression of *bsrbl* is modulated during the colonial blastogenic cycle [25]. The lectin can agglutinate microorganisms, both prokaryotic and eukaryotic, and foreign cells, thus rendering them more visible to immune cells. In addition, it acts as opsonin, promoting phagocytosis, and can activate both phagocytes, triggering their respiratory burst, and cytotoxic morula cells, stimulating their degranulation and the consequent induction of inflammation [25].

In the present study, we present our recent results on BsRBL. We focussed on its secretion mode and its subcellular distribution, using immunocytochemical analysis with both light and electron microscopy using a specific polyclonal antibody. In addition, by microinjecting the above antibody in the colonial circulation, we could demonstrate the influence of BsRBL in asexual reproduction and generation change.

## 2. Results

### 2.1. Specific Anti-BsRBL Antibodies Recognise Agglutinated Yeast Cells, B. schlosseri Phagocytes and Germ Cells

In agreement with previous observations [25], circulating phagocytes are the most abundant cell type recognised by the anti-BsRBL antibody. The incubation with pre-immune serum resulted in the absence of staining (Figure 1a). When not engaged in phagocytosis, phagocytes show a spreading morphology with a series of pseudopods sprouting from the cell body. In this case, labelling can be observed in the perinuclear region, in peripheral cytoplasmic regions or at the apex of pseudopodia (Figure 1b–d). When haemocytes were incubated with yeast cells, the latter frequently appear agglutinated: in this case, a clear labelling is visible in the interstices among cells (Figure 1e). In phagocytes that actively contact yeast cells, a strong labelling is associated with the contact region (plasma membrane and underlying cytoplasm) (Figure 1f,g). Upon the ingestion of yeast cells, phagocytes resorb their cytoplasmic projection, acquire a round morphology and show a strong labelling of the plasma membrane and the cytoplasm (Figure 1h,i). The intensity of the labelling was significantly increased (*p* < 0.05) in phagocytes previously exposed to yeast cells with respect to haemocytes incubated in filtered seawater (FSW; control) or treated only with the primary or the secondary antibody (Figure 2). The stain intensity at 5, 15, 30 and 60 min did not differ significantly, indicating that the maximum intensity of the immunolabelling is reached after only 5 min of exposure to nonself (Figure 2). In sexually mature colonies, the heads of the spermatozoa and the cytoplasm of oocytes resulted in being labelled by the antibody (Figure 1k,l).

Electron microscopy immunocytochemical analysis of yeast-incubated phagocytes show the absence of labelling and of labelled extracellular vesicles around cells incubated with a pre-immune serum (Figure 3a). Cell samples incubated with the anti-BsRBL antibody show the labelling of the phagocyte surface and of the cortical cytoplasm (Figure 3b,c), including the subtle projections connected with a thin stalk to the phagocytes (Figure 3e,f), sometimes completely disconnected from the cells (Figure 3b,d,g). Ingested yeast cells inside the phagosomes are abundantly labelled on their surfaces (Figure 3d).

### 2.2. Anti-BsRBL Antibody Recognises Extracellular Vesicles Released by Phagocytes

Although bioinformatics analyses predict the presence of a signal peptide corresponding to amino acids 1–21 [24], previous data [27] and the results of the present work suggest that the secretion of BsRBL can also occur in a non-canonical way, i.e., through the release of extracellular vesicles that, once in the medium, break up and discharge their content in the medium. To verify this, we obtained the extracellular vesicles from the haemocytes previously incubated with yeast cells (FSW in controls) by ultracentrifugation and subjected part of the pellet to immunocytochemical analysis with both light and electron microscopy. Data obtained at the light microscopy indicated that the extracellular vesicles aggregates from the yeast-exposed haemocytes are intensely labelled by the anti-BsRBL antibody, whereas those from the unexposed cells are much less stained (Figure 4a,b). In addition, in some cases, extracellular vesicles preparations contained some yeast cells that we were not able to eliminate completely: in these cases, yeast cells were adherent to extracellular vesicles (Figure 4a–c). Electron micrographs revealed that part of the extracellular vesicles were broken, probably as a consequence of ultracentrifugation, and their content formed a matrix in which the remaining vesicles were immersed. Few vesicles derived from unexposed haemocytes were labelled by the anti-BsRBL antibody and almost no labelling was observed in the matrix (Figure 4d). Conversely, a higher number of vesicles from yeast-exposed haemocytes were labelled and a diffuse labelling was observed in the surrounding matrix (Figure 4e). The occasional yeast cells found in the pellet after the ultracentrifugation showed a clear labelling of their surface (Figure 4f).

### 2.3. The Anti-BsRBL Antibody Slows the Progression of the Blastogenetic Cycle

When microinjected in the circulation of colonies approaching the TO, the anti-BsRBL antibody significantly (*p* < 0.05) increases the duration of the generation change at all the considered times, from 12 to 60 h, as demonstrated by the slower decrease of the size of the old, resorbing zooids with respect to control ones. For instance, after 24 and 36 h, when the new adult generation of control colonies opened its siphons, the old generation zooids were reduced to the 28% and 13% of their initial size, respectively, while in injected colonies, they were 66% and 40% of the original size, respectively (Figure 5a). Analogously, in BsRBL-injected colonies, the primary buds were always significantly (*p* < 0.05) smaller than the control ones. The delay in the bud growth rate disappeared only after 4 days from the completion of the generation change in control colonies (Figure 5b).

## 3. Discussion

Most human-secreted and cell surface proteins are post-translationally modified by glycosylation. The cell surface glycoproteins play a central role in modulating a variety of processes, such as signal transduction, cell–cell interactions and innate immunity [28,29,30]. The rising awareness of the importance of carbohydrates in cell biology has led to an increased effort to identify the actors involved in deciphering the glycocode residing in the glycans on cell surface glycoproteins and glycolipids [31,32]. Lectins are non-enzymatic glycan-binding proteins, widely diffuse in nature and involved in numerous biological processes, ranging from fertilisation and development to cell–cell interactions, signalling pathways and immune responses. As regards immunity, lectins can act as pattern recognition receptors as they sense and respond to changes in the sugar signature. They recognise nonself or altered self cells or molecules stimulating their clearance by phagocytosis or triggering an inflammatory response [33]. Among the many families of animal lectins [1,2,3,4,5], rhamnose-binding lectins form a family of proteins described in many animals, both vertebrates (fish) and invertebrates [10,11,12,13,14,15,16,17,18,19,20,21,22].

Ascidians are invertebrate chordates, members of the subphylum Tunicata, the latter considered the sister group of vertebrates [34]. Their peculiar phylogenetic position explains the interest towards various aspects of the biology of these organisms as they can help to elucidate some of the molecular and physiological events occurring in the course of the invertebrate–vertebrate transition. As invertebrates, ascidians rely on innate immunity, where sugars–lectin interaction play a pivotal role in the recognition of nonself [28].

In our previous studies on the actors involved in immune responses in the colonial ascidian *Botryllus schlosseri*, [24,25], we reported on the identification and the roles of a rhamnose-binding lectin that we called BsRBL. The lectin behaves as an opsonin enhancing the clearance, through phagocytosis, of microorganisms and activating the phagocyte respiratory burst. At higher concentrations, it can also trigger the degranulation of cytotoxic morula cells by activating an inflammatory cascade of reactions. Early studies [35] suggested that BsRBL is not released via the classical merocrine-like secretion. In the present work, we tried to elucidate this aspect by carrying out new immunocytochemical, and immunohistochemical labelling of colonies and of haemocytes using a specific polyclonal antibody raised against the purified BsRBL. Haemocytes were also challenged with yeast cells to stimulate their activation and induce phagocytosis. Light and electron microscopic analyses indicate that BsRBL is located on the surface of phagocytes. This fits the idea, supported by previous results [24,25,35] that the lectin behaves as opsonin, recognising, once released, both the phagocyte surface (the receptor on phagocytes is still unknown) and nonself cells and acting as a molecular bridge between them. Indeed, the observed staining of the contact regions among cells in agglutinated yeast and the observed immunopositivity of the yeast cell surface to the anti-BsRBL antibody that is maintained even after their ingestion by phagocytes, indicate that BsRBL can recognise the surface of yeast cells. The presence of the lectin in the cortical cytoplasm of the region of contact between phagocytes and yeast cells indicates a selective recruitment and release of the lectin in the contact region.

Immunolabelling, observable in the perinuclear and cortical cytoplasm of spreading phagocytes, as well as inside the cytoplasmic projections directed outwards and in the extracellular vesicles that frequently are seen around the phagocytes in the electron micrographs suggest that, in agreement with our previous suggestion [27], BsRBL is not released with a classic exocytosis. Although a signal peptide has been identified by bioinformatics algorithms [25], it seems that BsRBL can also be discharged via an alternative mode, i.e., through the formation of extracellular vesicles acting as cargoes. The abundance of immunostained extracellular vesicles obtained from the yeast-exposed cells as compared to the unstimulated cells fits this view. They probably release their content once they are detached from the cells upon their rupture, and we have no indications suggesting their possible re-cycling and re-entering the releasing cells by endocytosis. The adhesion of yeast cells to pelleted extracellular vesicles can be explained with the rupture of part of the vesicle, as a consequence of the effects of ultracentrifugation on the most fragile components of the pelleted material (there is no uniformity in the pelleted vesicles) and the consequent release of their content, BsRBL included. The latter links yeast cells to the packed vesicles.

According to our results, unstimulated spreading phagocytes carry out a constitutive synthesis of BsRBL, probably required for immunosurveillance. The synthesis is rapidly increased (within 5 min) upon the recognition of nonself and especially after the ingestion of foreign cells. This is likely mediated by cytotoxic morula cells that are the first cells sensing nonself in *Botryllus* and release phagocytosis-stimulating humoral factors (cytokines?) upon the recognition of foreign molecules [36,37,38].

BsRBL is present inside oocyte cytoplasm: this fits the numerous reports on echinoderms and fish, where it has been clearly shown that RBL is contained inside eggs, representing a sort of “internal defence” of the oocyte [12,13,18,20,39,40,41]. The distribution of BsRBL in the egg cytoplasm is quite uniform: this resembles what is observed in the unfertilised eggs of the echinoid *Anthocidaris crassispina*, where the rhamnose-binding lectin is stored in granules randomly distributed in the cytoplasm [42]. In the same species, upon fertilisation, the lectin migrate to the cortex from where it is released in the extracellular milieu to protect the developing embryo. This aspect has not been investigated in *Botryllus* and requires additional research.

As for the labelling of sperm cells, this is the first report showing the immunocytochemical labelling of spermatozoa. This observation fits the proposed role of rhamnose-binding lectins in fertilisation, in the interaction of the two gametes [15,16].

As stated in the introduction, *Botryllus* colonies undergo cyclical generation changes that characterise the colonial blastogenic cycle. We already demonstrated that the production and the release of BsRBL is modulated in the course of the blastogenic cycle, with a maximum production at the TO, when it is likely involved in the clearance of apoptotic cells and corpses by phagocytes [25]. During this period, the colony cannot eat as no siphons are open and it relies only on the recycling of nutrients deriving from the digestion of cells cleared by phagocytes [43,44]. Since: (i) BsRBL is involved in phagocytosis, (ii) phagocytosis is required for the progression of the blastogenic cycle [45] and (iii) there is evidence that effete cells change their surface carbohydrates [43], we used the anti-BsRBL antibody to interfere with the process of clearance of apoptotic cells and corpses. Indeed, as expected, the presence of the antibody significantly slows the resorption of the old zooids and the growth of the buds to adult size. As for the buds, in treated colonies, they reach adult size with a delay of 4 days with respect to the control colonies. It is conceivable, then, that buds from injected colonies suffer from a lack of adequate food during the critical period of time of the generation change but, once they have overcome this period, they grow to their adult size, even if with a delay with respect to the control colonies.

Collectively, our results contribute to a better understanding of the role of BsRBL in the biology of the *B. schlosseri* colonies. Future research will be directed to acquire better insights on the modulation of *bsrbl* transcription in the course of immune responses and on its role in fertilisation and early development.

## 4. Materials and Methods

### 4.1. Colonies of B. schlosseri

*B. schlosseri* is a colonial ascidian widely distributed in shallow waters all over the world. Colonies were collected in the Lagoon of Venice, near the marine station (Stazione Idrobiologica) of the University of Padova in Chioggia. They were transferred on 5 × 5 cm-glass slides and kept in aquaria in thermostated rooms at the Department of Biology, University of Padova, kept at a water temperature 19 °C, with 12:12 h of the light:dark cycle and fed every other day with unicellular algae of the genera *Dunaliella* and *Tetraselmis*.

### 4.2. Collection of Haemocytes and Phagocytosis Assay

Colonies, previously immersed for 5 min in a 0.38% sodium citrate in FSW to prevent haemocyte aggregation, were blotted dry and the peripheral vessel was punctured with a fine tungsten needle. Flowing haemolymph, collected with a glass micropipette, was transferred in a 1.5-mL vial at 4 °C. It was then centrifuged at 380× *g* for 10 min at 4 °C, and pelleted haemocytes were re-suspended in filtered seawater (FSW) to a final concentration of 5 × 10^5^ cells mL^−1^. Sixty µL of cell suspension were transferred on SuperFrost Plus (Menzel-Glaser, Braunschweig, Germany) glass slides, and cells were left to adhere for 30 min in a humid chamber. In the case of the phagocytosis assay, they were then incubated, for 1 h in FSW, in the presence or in the absence (control) of baker’s yeast (*Saccharomyces cerevisiae*) in FSW (yeast:haemocyte ratio = 10:1. Slides were then thoroughly washed by dipping them repeatedly in FSW to eliminate yeast cells and the haemocyte monolayers were fixed in 4% paraformaldehyde plus 0.1% glutaraldehyde in 0.4 M cacodylate buffer, pH 7.4.

### 4.3. Immunocytochemistry and Immunohistochemistry

Polyclonal antibody against BsRBL was obtained as already described [25]. Haemocytes were incubated with yeast (FSW in controls) for 5, 15, 30 and 60 min. They were then fixed as described above, washed in a phosphate-buffered saline (PBS: 1.37 M NaCl, 0.03 M KCl, 0.015 M KH_2_PO_4_, 0.065 M Na_2_HPO_4_, pH 7), permeabilised by a 10-min incubation in 0.1% Triton X-100 in PBS, incubated for 30 min in 1% H_2_O_2_ in methanol, to block endogenous peroxidase, and treated for 30 min with 3% powdered milk in PBS to prevent unspecific binding. Haemocytes were then incubated overnight in a rabbit polyclonal anti-BsRBL antibody [25] diluted 1/1000 in PBS (rabbit preimmune serum in controls), followed by washing in PBS and incubation for 2 h in biotinylated goat anti-rabbit IgG antibody (Jackson Immunoresearch Europe Ltd., Ely, UK) diluted 1/2000 in PBS. After an additional washing in PBS, cells were incubated in an avidin-biotin-peroxidase complex (ABC; Vectastain, Vector Laboratories, Burlingame, CA, USA) for 30 min, washed again in PBS and incubated for 5 min in 0.63 mM 3,3′-diaminobenzidine (DAB) plus 4% H_2_O_2_ in PBS. Slides were then mounted with Acquovitrex (Carlo Erba, Cornaredo, Italy) and observed under the light microscope.

For immunohistochemical analysis, colonies were fixed for 60 min in 4% paraformaldehyde plus 0.1% glutaraldehyde in 0.4 M cacodylate buffer, pH 7.4, dehydrated and embedded in Paraplast Xtra (McCormick Scientific, St Louis, MO, USA) and cut to 7-µm sections with a Leitz 1212 microtome (Leitz, Wetzlar, Germany). Sections were left to adhere to SuperFrost Plus (Menzel-Glaser, Braunschweig, Germany) glass slides and, once dewaxed, they were incubated overnight in anti-BsRBL antibody, followed by washing in PBS and incubation for 30 min in 1% H_2_O_2_ in methanol, to block endogenous peroxidase, and a 30-min treatment with 3% powdered milk in PBS, to prevent unspecific binding. They were then incubated overnight in the anti-BsRBL primary antibody at the dilution reported above, washed in PBS and incubated for 2 h in goat anti-rabbit IgG secondary antibody. Then, the slides were treated with ABC followed by 20 min in DAB and H_2_O_2_ as previously described, dehydrated and mounted in Eukitt (ORSArtec GmbH, Bobingen, Germany). In both cases, positive sites were stained dark brown. The intensity of the labelling, in terms of the “black value” of the pixels, was calculated on images acquired with the Infinity Capture software (release 5.0, Teledyne Lumenera, Ottawa, ON, Canada) and analysed using the ImageJ software (release 2.0.0-beta-7, https://imagej.nih.gov.ij, accessed on 6 January 2023). Briefly, once the contour of the phagocytes was traced with the specific software tool, the “measure” function returns the “mean grey value” of the pixels contained within the selected area. This value is proportional to the staining intensity of the precipitate deriving from the oxidation of DAB and represents an indirect estimate of the concentration of BsRBL on the cells. All pictures were acquired using the same microscope brightness settings.

### 4.4. Extracellular Vesicles Isolation and Characterisation

Haemolymph, harvested as described before from large colonies (more than 2000 zooids in size), was centrifuged, as described before, and the pelleted haemocytes were re-suspended in FSW to reach a final concentration of 10^6^ cells mL^−1^. Part of them were incubated for 10 min with yeast (yeast:haemocyte ratio = 10:1), and part in FSW (control). To get extracellular vesicles, we followed the procedure described in Lobb et al. [46]. Briefly, after the incubation, the cell suspensions were centrifuged at 300× *g* for 10 min at 4 °C. Supernatants were then collected, diluted to 30 mL of FSW in order to fill the ultracentrifuge tubes and filtered through 0.22-μm filters. The suspensions were then centrifuged at 100,000× *g* at 4 °C for 90 min. Supernatants were then removed and pellets were resuspended in 30 mL of ice-cold FSW. A second round of ultracentrifugation was carried out with the same conditions.

For light microscope immunocytochemical analysis, ultracentrifuged pellets were partly resuspended in 1 mL of FSW and then left to adhere for 48 h on SuperFrost Plus (Menzel-Glaser) glass slides in a humid chamber. They were then fixed for 30 min at 4 °C in 4% paraformaldehyde and 0.1% glutaraldehyde in 0.4 M cacodylate buffer, pH 7.4), washed in PBS for 30 min and permeabilised with 0.1% Triton™ X-100 in PBS for 10 min. Slides were then incubated in 1% H_2_O_2_ in methanol for 30 min, washed in PBS and then incubated for 40 min in 5% powdered milk in PBS. Samples were then incubated overnight at 4 °C in an anti-BsRBL primary antibody diluted 1:1000 in PBS (rabbit preimmune serum in controls) and, after washing in PBS for 2 h with goat anti-rabbit IgG secondary antibody (Jackson Immunoresearch Europe Ltd., Ely, UK), was diluted as reported before. After a wash in PBS, they were incubated in ABC (Vectastain, Vector Laboratories, Burlingame, CA, USA) for 30 min and finally treated with 0.63 mM DAB plus 4% H_2_O_2_ in PBS for 10 min. Slides were finally mounted with Acquovitrex (Carlo Erba, Cornaredo, Italy) and observed under the light microscope.

For electron microscopy, pelleted extracellular vesicles were fixed overnight in 4% paraformaldehyde plus 0.1% glutaraldehyde in 0.4 M cacodylate buffer, pH 7.4. Samples were then embedded in LRWhite (Sigma-Aldrich, St. Louis, MO, USA). Ultrathin sections (100 nm) were collected on copper grids, washed in PBS for 15 min, treated with 5% powdered milk in PBS for 40 min and then incubated overnight with the anti-BsRBL primary antibody, diluted as reported above. Sections were then washed in PBS and incubated for 2 h in goat anti-rabbit IgG secondary antibody conjugated with 15-nm colloidal gold particles (British Biocell International, Freiburg, Germany). Finally, sections were counterstained with uranyl acetate prior to examination under the electron microscope.

### 4.5. Effects of the Anti-BsRBL Antibody on the Blastogenetic Cycle

Colonies were split in two subclones of 2–3 systems each. When approaching (one day before) the TO, one of the subclones was microinjected with 5 µL of anti-BsRBL antibody, diluted 1/250 in FSW. As controls, untreated subclones or subclones microinjected with 5 µL of FSW or preimmune serum were used. Microinjections were repeated every 12 h until the completion of the TO. The surfaces of the zooids and buds were measured with ImageJ software (release 2.0.0-beta-7, http://imagej.nih.gov.ij, accessed on 6 January 2023).

### 4.6. Statistical Analysis

All the experiments were carried out in triplicate with three different colonies (*n* = 3). Data were analysed with the one-way ANOVA (Tukey’s test) associated with linear regression analysis, followed by Duncan’s test for mean comparisons. Differences were considered statistically significant when *p* < 0.05.

## 5. Conclusions

In the present work, we confirmed that BsRBL is secreted by circulating phagocytes, mainly through the production of extracellular vesicles that, by rupturing, release the lectin in the surrounding medium.

Its synthesis and release are triggered by the recognition of foreign molecules or cells, and it recognises and binds the surface of microbial cells forming a molecular bridge with the phagocyte surface, thus promoting the clearance, by phagocytosis, on the nonself material.

The observed interference of the specific antibody on the correct completion of the generation change is an additional confirmation of the importance of phagocytosis in the process of TO, as the clearance of apoptotic cells is required for the progression of the process.

The labelling of egg cytoplasm and sperm heads open new questions on the role of BsRBL in fertilisation that will be the subject of future research.

## Figures and Tables

**Figure 1 marinedrugs-21-00171-f001:**
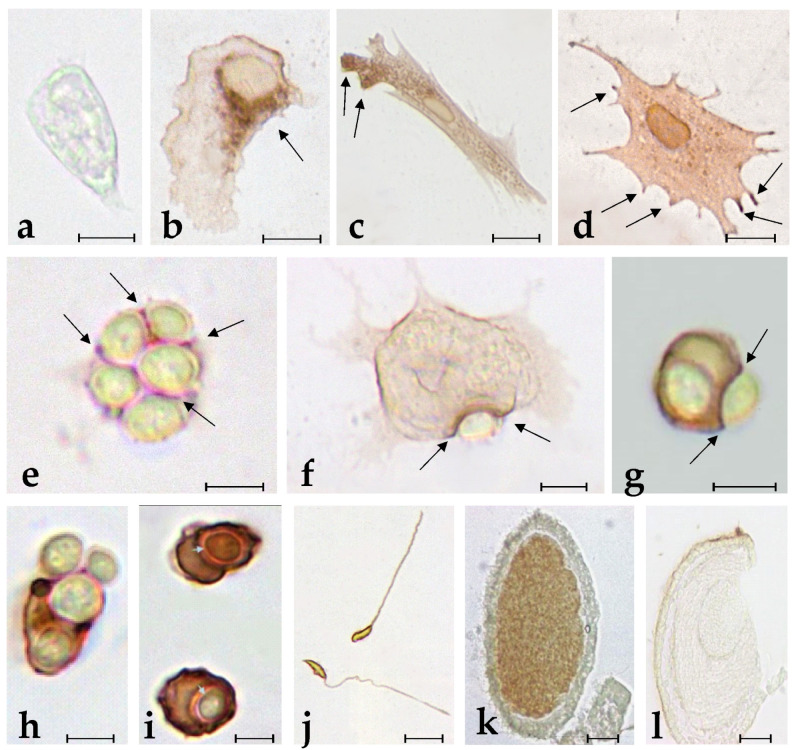
Immunolabelling of *B. schlosseri* cells with anti-BsRBL antibody. (**a**) Control phagocyte incubated with preimmune serum; (**b**–**d**) Spreading circulating phagocytes showing immunopositivity (arrows) in the perinuclear region (**b**), in the peripheral cytoplasm (**c**) and inside the cell projections (**d**); (**e**) Agglutinated yeast cells in the presence of BsRBL: the interstices among cells are labelled by the antibody (arrows); (**f**–**h**) Phagocytes contacting yeast cells showing immunopositivity in the contact region (arrows in **f**,**g**) and in the cytoplasm where other yeast cells have been ingested (**g**,**h**); (**i**) Round phagocytes with ingested yeast cells and an intensely labelled cytoplasm; (**j**) Spermatozoa with immunopositivity in their heads; (**k**,**l**) *B. schlosseri* eggs: the specimen in (**k**) was incubated in the primary antibody that was omitted in (**l**). Scale bar: 5 µm in (**a**–**j**), 50 µm in (**k**,**l**).

**Figure 2 marinedrugs-21-00171-f002:**
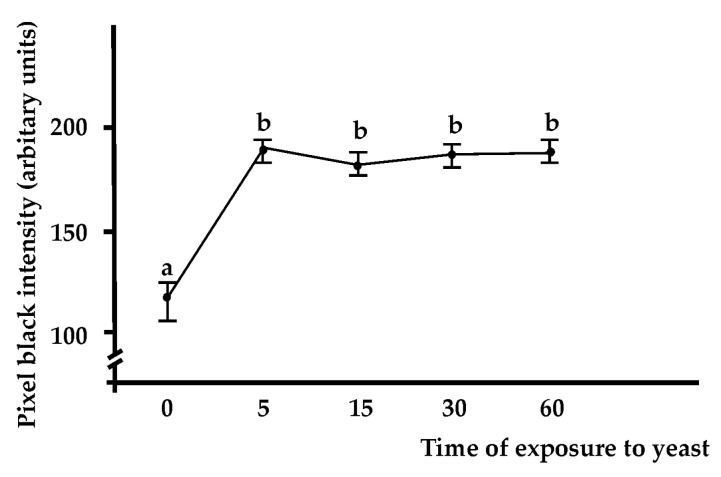
Black intensity of the pixels in phagocytes immunostained with anti-BsRBL antibody. The cells were incubated with yeast cells for various periods of times (0: control cells incubated in FSW). Different letters mark significant (*p* < 0.05) differences.

**Figure 3 marinedrugs-21-00171-f003:**
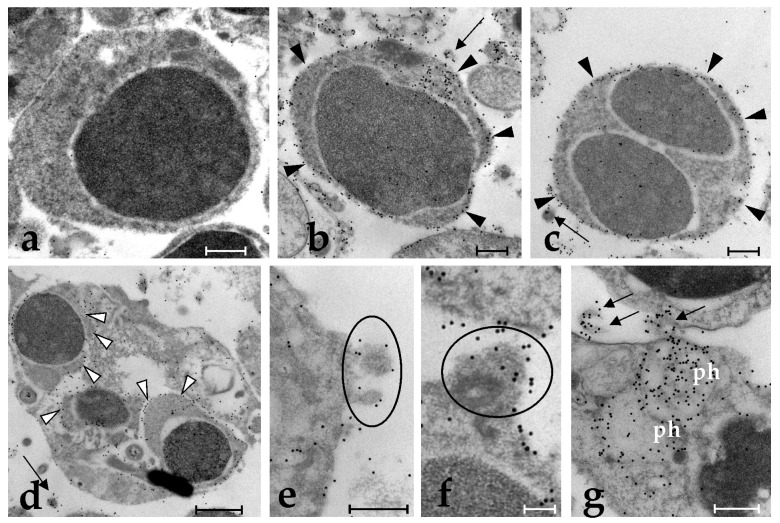
Immunolabeling of *B. schlosseri* circulating cells with anti-BsRBL antibody, electron microscopic analysis: (**a**) Control phagocyte incubated with preimmune serum; (**b**,**c**) Circulating phagocytes showing immunopositivity on their surface (arrowheads); (**d**) Phagocyte with ingested yeast cells, the latter showing immunopositivity along their contour (arrowheads); a stained extracellular vesicle is visible (arrow); (**e**,**f**) Phagocyte details showing immunopositive cytoplasmic projections (encircled); (**g**) Detail of a phagocyte. Marked extracellular vesicles are indicated by arrows; ph: phagosomes. Scale bars: 0.5 µm in (**a**–**c**,**e**,**g**); 1 µm in (**d**); 0.1 µm in (**f**).

**Figure 4 marinedrugs-21-00171-f004:**
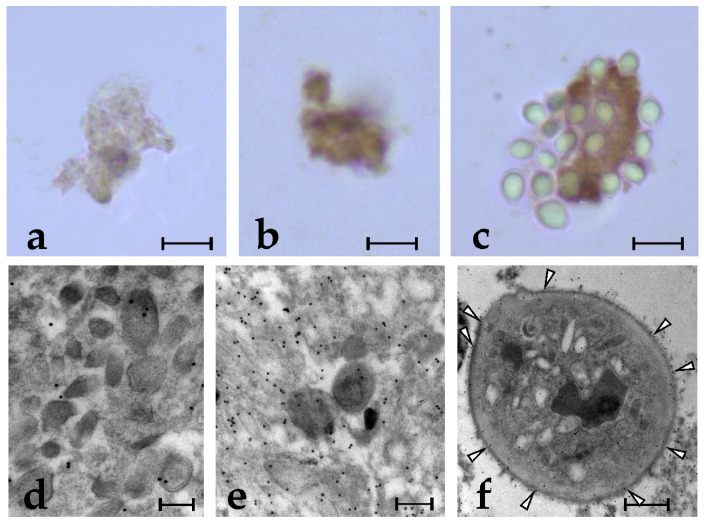
(**a**–**f**) Pellets of extracellular vesicles from *B. schlosseri* haemocytes stained immunocytochemically with the anti-BsRBL antibody and observed under the light (**a**–**c**) and the electron (**d**–**f**) microscope. (**a**,**d**) Vesicles from cells exposed to FSW (control); (**b**,**e**) Vesicles from cells exposed to yeast; (**c**) Vesicles from the previous source showing many adherent yeast cells; (**f**) Yeast cell labelled along its surface (arrowheads). Scale bars: 5 µm in (**a**–**c**), 0.2 µm in (**d**,**e**), 1 µm in (**f**).

**Figure 5 marinedrugs-21-00171-f005:**
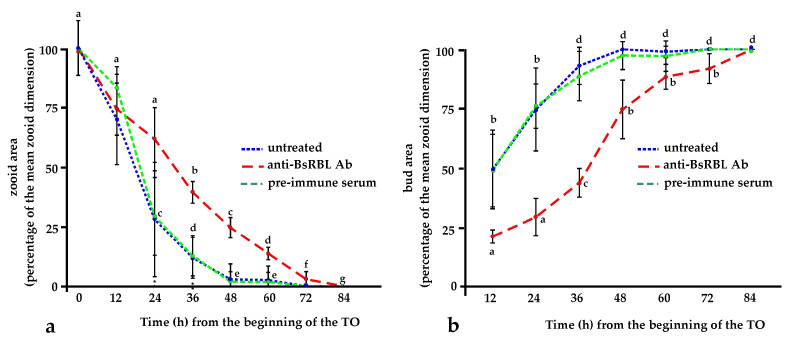
(**a**) Relative size (area) of resorbing zooids, expressed as a percentage of the mean zooid size, during the TO in untreated colonies (blue line) and in colonies injected with preimmune serum (control; green line) or BsRBL (red line); (**b**) Relative size (area) of maturing primary buds, expressed as a percentage of the mean zooid size, during the TO in untreated colonies (blue line) and in colonies injected with preimmune serum (control; green line) or BsRBL (red line). Different letters mark significant differences (*p* < 0.05).

## Data Availability

All data generated or analysed during this study are included in this published article.

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
