# Peer review of "New Data on the Rhamnose-Binding Lectin from the Colonial Ascidian Botryllus schlosseri: Subcellular Distribution, Secretion Mode and Effects on the Cyclical Generation Change"

_marinedrugs, 2023, doi:10.3390/md21030171_

Round 1

Reviewer 1 Report

The manuscript, “New data on the rhamnose-binding lectin from the colonial ascidian Botryllus schlosseri: tissue distribution, secretion mode and effects on the cyclical generation change” by Bovo et al. presents recent results on B. schlosseri rhamnose-binding lectin (BsRBL). Also, this study focused on its secretion mode and its distribution in tissues, using immunocytochemical analysis at both light and electron microscopy. In addition, using specific polyclonal antibodies, it was demonstrated the influence of BsRBL in asexual reproduction and generation changes.

Comments and Suggestions for Authors:

1- Lines 9-21 (Abstract): The abstract part is more like the introduction. For example, the method material, results, and conclusions of this study are not mentioned.

2- Line 22: Correct the keywords according to MeSH.

3-Lines 36-38: Add the full form of “BsRBL”. (B. schlosseri rhamnose-binding lectin)

4- Lines 213-218: In this section, mention the temperature of the aquarium water and the feeding schedule.

5- Line 263-264, 306-307: In this section, it is only said that ImageJ software is used, please provide more details about the use of ImageJ.

6- Add a conclusion paragraph at the end of the article.

Author Response

We thanks the reviewer for his/her suggestions useful to improve the manuscript. Here below our replies.

Answer to reviewer 1

1- Lines 9-21 (Abstract): The abstract part is more like the introduction. For example, the method material, results, and conclusions of this study are not mentioned.

done

2- Line 22: Correct the keywords according to MeSH.

done

3-Lines 36-38: Add the full form of “BsRBL”. (B. schlosseri rhamnose-binding lectin)

done

4- Lines 213-218: In this section, mention the temperature of the aquarium water and the feeding schedule.

done

5- Line 263-264, 306-307: In this section, it is only said that ImageJ software is used, please provide more details about the use of ImageJ.

added

6- Add a conclusion paragraph at the end of the article.

added

Reviewer 2 Report

Dear  Editor , the manuscript entitled : "New data on the rhamnose-binding lectin from the colonial ascidian Botryllus schlosseri: tissue distribution, secretion mode and effects on the cyclical generation change" by Bovo and Ballarin describes an update about the tissue distribution and the mechanisms of secretion of a rhamnose-binding lectin from Botryllus schlosseri already characterized in the past by the same group.

The study is interesting and for sure relevant for this journal.

Regarding to the Figures (in particular Figures 1 and 3) I will suggest adding an additional panel with the negative controls for the circulating phagocytes.

Indeed, please check the location of the statistical data within the text. For instance, line 70 describes a statistical significance that is not reported within figure 2.  Please check these data and , eventually, review the figure legends

Author Response

We thank the anonymous reviewer for the suggestions useful to imprve the manuscript. Here below our replies.

Answer to reviewer 2

Regarding to the Figures (in particular Figures 1 and 3) I will suggest adding an additional panel with the negative controls for the circulating phagocytes.

added

Indeed, please check the location of the statistical data within the text. For instance, line 70 describes a statistical significance that is not reported within figure 2.  Please check these data and , eventually, review the figure legends

the text was checked and modified where necessary, according to the suggestion.

Reviewer 3 Report

In this study, the authors studied the distribution of a kind of lectin that their lab. identified previously by a special antibody. More interesting is that they found that the presence of this antibody significantly slowed the cyclical generation in colony ascidian. I think the current version might reach the quality in Marine Drugs given the following points could be solved.

1.     In the title: New data on the rhamnose-binding lectin from the colonial ascidian Botryllus schlosseri: tissue distribution, secretion mode, and effects on the cyclical generation change. It might be better to change the “tissue distribution” to “subcellular distribution”,which was investigated in this work.

2.     In Figure 3a,b, Electron microscopy analysis showed the distribution of BsRBL in the phagocyte surface, Does this mean it localize in the membrane or membrane-bind? Detailed interpretation is needed. 

3.     Line 107-108

If BsRBL secretion occurs via an apocrine-like secretion, it means that the vesicles detach from the apical pole of the cell, please give detailed explanation of phagocyte polarity.

Is it possible that BsRBL secretion via both canonical and non-canonical way Since the existence of signal peptide and the presentence of soluble extracellular BsRBL? It seems that the evidences for pointing to a non-canonical way are not sufficient. 

4.     Line 114-118: 

The author said that the extracellular vesicles preparations contained some yeast cells adherent to them. But since the BsRBL was enclosed by the vesicle membrane, what is mechanism driving the contact? 

The authors stated that parts of the extracellular vesicles were broken and their content formed a matrix in which the remaining vesicles were immersed. So, why did some vesicles break down to release BsRBL and some keep intact? The intact vesicles will further enter cells by endocytosis?  

5.     The authors stated that neutralization of the rhamnose-binding lectins could slow the progression of the blastogenetic cycle by inhibition the clearance of the old cell, which means that without anti-BsRBL antibody the BsRBL could recognize the old tissue cells of the resorbing zooids, and then ingest by the phagocytes to provide nutrients for the bud growing. So, do the old cells be marked by specific carbohydrate such as rhamnose that make them different from the normal tissue cells?

Additional evidences are needed to confirm the current statement. Is it possible for the authors to add more lectin in ascidian to see whether the blastogenetic cycle is changed?

Author Response

Thank to the anonymous referee for his/her useful suggestions that helped us to improve the manuscript.

Here below our replies to his/her comments

Answer to reviewer 3

  1. In the title: New data on the rhamnose-binding lectin from the colonial ascidian Botryllus schlosseri: tissue distribution, secretion mode, and effects on the cyclical generation change. It might be better to change the “tissue distribution” to “subcellular distribution”,which was investigated in this work.

Thanks for the suggestion. We modified the title.

  1. In Figure 3a,b, Electron microscopy analysis showed the distribution of BsRBL in the phagocyte surface, Does this mean it localize in the membrane or membrane-bind? Detailed interpretation is needed.

We tried to explain it better in the text. There are no indications of a membrane form of the lectin (no transmembrane domain). According to the literature, it binds to some sugars on the cell surface, the nature of which, in Botryllus, is still unknown

  1. Line 107-108

If BsRBL secretion occurs via an apocrine-like secretion, it means that the vesicles detach from the apical pole of the cell, please give detailed explanation of phagocyte polarity.

With the term “apocrine-like” we intended a secretion via the release of vesicles, without suggesting a polarity of the cell. We deleted the term and simply explained that BsRBL can be released also via this non-canonical secretion way.

Is it possible that BsRBL secretion via both canonical and non-canonical way. Since the existence of signal peptide and the presentence of soluble extracellular BsRBL? It seems that the evidences for pointing to a non-canonical way are not sufficient.

The reviewer is right. We have no indications to exclude the canonical pathway. The text was changed accordingly.

  1. Line 114-118:

The author said that the extracellular vesicles preparations contained some yeast cells adherent to them. But since the BsRBL was enclosed by the vesicle membrane, what is mechanism driving the contact?

As explained in the text, we hypothesise that the vesicles rupture once detached from the cells thus releasing their contents, BsRBL included.

The authors stated that parts of the extracellular vesicles were broken and their content formed a matrix in which the remaining vesicles were immersed. So, why did some vesicles break down to release BsRBL and some keep intact? The intact vesicles will further enter cells by endocytosis? 

This likely depends on ultracentrifugation which damages the most fragile vesicles (there is no uniformity in the pelleted vesicles)

  1. The authors stated that neutralization of the rhamnose-binding lectins could slow the progression of the blastogenetic cycle by inhibition the clearance of the old cell, which means that without anti-BsRBL antibody the BsRBL could recognize the old tissue cells of the resorbing zooids, and then ingest by the phagocytes to provide nutrients for the bud growing. So, do the old cells be marked by specific carbohydrate such as rhamnose that make them different from the normal tissue cells?

Yes, old cells change their surface carbohydrates. This was explained better in the text and derives from previous published research (Cima et al. Hovering between death and life: haemocytes and natural apoptosis in the blastogenetic cycle of the colonial ascidian Botryllus schlosseri. Dev. Comp. Immunol. 2010, 34, 272-285. doi: 10.1016/j.dci.2009.10.005)

Additional evidences are needed to confirm the current statement. Is it possible for the authors to add more lectin in ascidian to see whether the blastogenetic cycle is changed?

It has been already demonstrated that impairments of phagocytosis led to a slowing of the cycle (Voskoboynik, et al. Macrophage involvement for successful degeneration of apoptotic organs in the colonial urochordate Botryllus schlosseri. J. Exp. Biol. 2004, 207, 2409-2416. doi: 10.1242/jeb.01045). Our attempts to inject lectins in the past were not successful as they, at the concentrations used (the same concentration to reveal sugars in cells in cytological analyses), were toxic for the colony. Probably, we could re-start this investigation in the future with lower doses.

Round 2

Reviewer 2 Report

Dear Editor,

the Authors introduced the requested changes therefore, I suggest the pubblication of the manuscript in this new revised form.

Reviewer 3 Report

I am satisfied with the revision version and have no further question.